# ONLINE META-CRITIC LEARNING FOR OFF-POLICY ACTOR-CRITIC METHODS

## ABSTRACT

Off-Policy Actor-Critic (Off-PAC) methods have proven successful in a variety of continuous control tasks. Normally, the critic's action-value function is updated using temporal-difference, and the critic in turn provides a loss for the actor that trains it to take actions with higher expected return. In this paper, we introduce a novel and flexible *meta*-critic that observes the learning process and meta-learns an additional loss for the actor that accelerates and improves actor-critic learning. Compared to the vanilla critic, the meta-critic network is explicitly trained to accelerate the learning process; and compared to existing meta-learning algorithms, meta-critic is rapidly learned online for a single task, rather than slowly over a family of tasks. Crucially, our meta-critic framework is designed for off-policy based learners, which currently provide state-of-the-art reinforcement learning sample efficiency. We demonstrate that online meta-critic learning leads to improvements in a variety of continuous control environments when combined with contemporary Off-PAC methods DDPG, TD3 and the state-of-the-art SAC.

## 1 INTRODUCTION

Off-policy Actor-Critic (Off-PAC) methods are currently central in deep reinforcement learning (RL) research due to their greater sample efficiency compared to on-policy alternatives. On-policy requires new trajectories to be collected for each update to the policy, and is expensive as the number of gradient steps and samples per step increases with task-complexity even for contemporary TRPO (Schulman et al., 2015), PPO (Schulman et al., 2017) and A3C (Mnih et al., 2016) algorithms. Off-policy methods, such as DDPG (Lillicrap et al., 2016), TD3 (Fujimoto et al., 2018) and SAC (Haarnoja et al., 2018b) achieve greater sample efficiency due to their ability to learn from randomly sampled historical transitions without a time sequence requirement, thus making better use of past experience. Their critic estimates the action-value (Q-value) function using a differentiable function approximator, and the actor updates its policy parameters in the direction of the approximate action-value gradient. Briefly, the critic provides a loss to guide the actor, and is trained in turn to estimate the environmental action-value under the current policy via temporal-difference learning (Sutton et al., 2009). In all these cases the learning algorithm itself is hand-crafted and fixed.

Recently meta-learning, or "learning-to-learn" has become topical as a paradigm to accelerate RL by learning aspects of the learning strategy, for example, through learning fast adaptation strategies (Finn et al., 2017; Rakelly et al., 2019; Riemer et al., 2019), exploration strategies (Gupta et al., 2018), optimization strategies (Duan et al., 2016b), losses (Houthooft et al., 2018), hyperparameters (Xu et al., 2018; Veeriah et al., 2019), and intrinsic rewards (Zheng et al., 2018). However, the majority of these works perform meta-learning on a family of tasks or environments and amortize this huge cost by deploying the trained strategy for fast learning on a new task.

In this paper we introduce a novel meta-critic network to enhance existing Off-PAC learning frameworks. The meta-critic is used alongside the vanilla critic to provide a loss to guide the actor's learning. However compared to the vanilla critic, the meta-critic is explicitly (meta)-trained to accelerate the learning process rather than merely estimate the action-value function. Overall, the actor is trained by gradients provided by both critic and meta-critic losses, the critic is trained by temporal-difference as usual, and the meta-critic is trained to generate maximum learning performance improvements in the actor. In our framework, both the critic and meta-critic use randomly sampled off-policy transitions for efficient and effective Off-PAC learning, providing superior sam-

ple efficiency compared to existing on-policy meta-learners. Furthermore, we demonstrate that our meta-critic can be successfully learned *online* within a single task. This is in contrast to the currently widely used meta-learning research paradigm – where entire task *families* are required to provide enough data for meta-learning, and to provide new tasks to amortize the huge cost of meta-learning.

Essentially our framework meta-learns an auxiliary loss function, which can be seen as an intrinsic motivation towards optimum learning progress (Oudeyer & Kaplan, 2009). As analogously observed in several recent meta-learning studies (Franceschi et al., 2018), our loss-learning can be formalized as a bi-level optimization problem with the upper level being meta-critic learning, and lower level being conventional learning. We solve this joint optimization by iteratively updating the meta-critic and base learner online while solving a single task. Our strategy is thus related to the meta-loss learning in EPG (Houthooft et al., 2018), but learned online rather than offline, and integrated with Off-PAC rather than their on-policy policy-gradient learning. The most related prior work is LIRPG (Zheng et al., 2018), which meta-learns an intrinsic reward online. However, their intrinsic reward just provides a helpful scalar offset to the environmental reward for on-policy trajectory optimization via policy-gradient (Sutton et al., 2000). In contrast our meta-critic provides a loss for direct actor optimization just based on sampled transitions, and thus achieves dramatically better sample efficiency than LIRPG reward learning in practice. We evaluate our framework on several contemporary continuous control benchmarks and demonstrate that online meta-critic learning can be integrated with and improve a selection of contemporary Off-PAC algorithms including DDPG, TD3 and SAC.

## 2 BACKGROUND AND RELATED WORK

**Policy-Gradient (PG) Methods.** *On-policy* methods usually update actor parameters in the direction of greater cumulative reward. However, on-policy methods need to interact with the environment in a sequential manner to accumulate rewards and the expected reward is generally not differentiable due to environment dynamics. Even exploiting tricks like importance sampling and improved application of A2C (Zheng et al., 2018), the use of full trajectories is less effective than off-policy transitions, as the trajectory needs a series of continuous transitions in time. Off-policy actor-critic architectures aim to provide greater sample efficiency by reusing past experience (previously collected transitions). DDPG (Lillicrap et al., 2016) borrows two main ideas from Deep Q Networks (Mnih et al., 2013; 2015): a big replay buffer and a target Q network to give consistent targets during temporal-difference backups. TD3 (Twin Delayed Deep Deterministic policy gradient algorithm) (Fujimoto et al., 2018) develops a variant of Double Q-learning by taking the minimum value between a pair of critics to limit over-estimation. SAC (Soft Actor-Critic) (Haarnoja et al., 2018a;b) proposes a maximum entropy RL framework where its stochastic actor aims to simultaneously maximize expected action-value and entropy. The latest version of SAC (Haarnoja et al., 2018b) also includes the "the minimum value between both critics" idea in its implementation.

**Meta Learning for RL.** Meta-learning (a.k.a. learning to learn) (Santoro et al., 2016; Finn et al., 2017) has received a resurgence in interest recently due to its potential to improve learning performance, and especially sample-efficiency in RL (Gupta et al., 2018). Several studies learn optimizers that provide policy updates with respect to known loss or reward functions (Andrychowicz et al., 2016; Duan et al., 2016b; Meier et al., 2018). A few studies learn hyperparameters (Xu et al., 2018; Veeriah et al., 2019), loss functions (Houthooft et al., 2018; Sung et al., 2017) or rewards (Zheng et al., 2018) that steer the learning of standard optimizers. Our meta-critic framework is in the category of loss-function meta-learning, but unlike most of these we are able to meta-learn the loss function online in parallel to learning a single extrinsic task rather. No costly offline learning on a task family is required as in Houthooft et al. (2018); Sung et al. (2017). Most current Meta-RL methods are based on on-policy policy-gradient, limiting their sample efficiency. For example, while LIRPG (Zheng et al., 2018) is one of the rare prior works to attempt online meta-learning, it is ineffective in practice due to only providing a scalar reward increment rather than a loss for direct optimization. A few meta-RL studies have begun to address off-policy RL, for conventional offline multi-task meta-learning (Rakelly et al., 2019) and for optimising transfer vs forgetting in continual learning of multiple tasks (Riemer et al., 2019). The contribution of our Meta-Critic is to enhance state-of-the-art Off-PAC RL with single-task online meta-learning.

**Loss Learning.** Loss learning has been exploited in 'learning to teach' (Wu et al., 2018) and surrogate loss learning (Huang et al., 2019; Grabocka et al., 2019) where a teacher network predicts the

parameters of a manually designed loss in supervised learning. In contrast our meta-critic is itself a differentiable loss, and is designed for use in reinforcement learning. Other applications learn losses that improve model robustness to out of distribution samples (Li et al., 2019; Balaji et al., 2018). Our loss learning architecture is related to Li et al. (2019), but designed for accelerating single-task Off-PAC RL rather than improving robustness in multi-domain supervised learning.

## 3 METHODOLOGY

We aim to learn a meta-critic that provides an auxiliary loss $L_\omega^{\text{aux}}$ to assist the actor's learning of a task. The auxiliary loss parameters $\omega$ are optimized in a meta-learning process. The vanilla critic $L^{\text{main}}$ and meta-critic $L_\omega^{\text{aux}}$ losses train the actor $\pi_\phi$ off-policy via stochastic gradient descent.

### 3.1 REVIEW OF OFF-POLICY ACTOR-CRITIC RL

Reinforcement learning involves an agent interacting with the environment $E$. At each time $t$, the agent receives an observation $s_t$, takes a (possibly stochastic) action $a_t$ based on its policy $\pi : \mathcal{S} \to \mathcal{A}$, and receives a scalar reward $r_t$ and new state of the environment $s_{t+1}$. We call $(s_t, a_t, r_t, s_{t+1})$ as a single point transition. The objective of RL is to find the optimal policy $\pi_\phi$, which maximizes the expected cumulative return $J$.

In on-policy RL, $J$ is defined as the discounted episodic return based on a sequential trajectory over horizon $H$: $(s_0, a_0, r_0, \cdots, s_H, a_H, r_H)$. $J = \mathbb{E}_{r_t, s_t \sim E, a_t \sim \pi} \left[ \sum_{t=0}^{H} \gamma^t r_t \right]$. In the usual implementation of A2C, $r$ is represented by a surrogate state-value $V(s_t)$ from its critic. Since $J$ is only a scalar value, the gradient of $J$ with respect to policy parameters $\phi$ has to be optimized under the policy gradient theorem (Sutton et al., 2000): $\nabla_\phi J(\phi) = \mathbb{E} \left[ J \nabla_\phi log \, \pi_\phi(a_t | s_t) \right]$.

In off-policy RL (e.g., DDPG, TD3, SAC) which is our focus in this paper, parameterized policies $\pi_\phi$ can be directly updated by defining the actor loss in terms of the expected return $J(\phi)$ and taking its gradient $\nabla_\phi J(\phi)$, where $J(\phi)$ depends on the action-value $Q_\theta(s_t, a_t)$. The main loss $L^{\text{main}}$ provided by the vanilla critic is thus

$$L^{\text{main}} = -J(\phi) = -\mathbb{E}_{s \sim p_\pi} Q_\theta(s, a)|_{a = \pi_\phi(s)}, \tag{1}$$

where we follow the notation in TD3 and SAC that $\phi$ and $\theta$ denote actors and critics respectively.

The main loss is calculated by a mini-batch of transitions randomly sampled from the replay buffer. The actor's policy network is updated as $\Delta\phi = \alpha \nabla_\phi L^{\text{main}}$, following the critic's gradient to increase the likelihood of actions that achieve a higher Q-value. Meanwhile, the critic uses Q-learning updates to estimate the action-value function:

$$\theta \leftarrow \arg\min_\theta \left( Q_\theta(s_t, a_t) - r_t - \gamma Q_\theta(s_{t+1}, \pi(s_{t+1})) \right)^2. \tag{2}$$

### 3.2 ALGORITHM OVERVIEW

Our meta-learning goal is to train an auxiliary meta-critic network $L_\omega^{\text{aux}}$ that in turn enhances actor learning. Specifically, it should lead to the actor $\phi$ having improved performance on the main task $L^{\text{main}}$ when following gradients provided by the meta-critic as well as those provided by the main task. This can be seen as a bi-level optimization problem (Franceschi et al., 2018; Rajeswaran et al., 2019) of the form:

$$\omega = \underset{\omega}{\text{argmin}} \, L^{\text{meta}}(d_{val}; \phi^*)$$
$$s.t. \ \ \phi^* = \underset{\phi}{\text{argmin}} \left( L^{\text{main}}(d_{trn}; \phi) + L_\omega^{\text{aux}}(d_{trn}; \phi) \right), \tag{3}$$

where we can assume $L^{\text{meta}}(\cdot) = L^{\text{main}}(\cdot)$ for now. Here the lower-level optimization trains the actor $\phi$ to minimize both the main task and meta-critic-provided losses on some training samples. The upper-level optimization further requires the meta-critic $\omega$ to have produced a learned actor $\phi^*$ that minimizes a meta-loss that measures the actor's main task performance on a second set of validation

---

**Algorithm 1** Online Meta-Critic Learning for Off-PAC RL

---

$\phi, \theta, \omega, \mathcal{D} \leftarrow \emptyset$      // Initialize actor, critic, meta-critic and buffer
**for** each iteration **do**
    **for** each environment step **do**
        $a_t \sim \pi_\phi(a_t | s_t)$      // Select action according to the current policy
        $s_{t+1} \sim p(s_{t+1} | s_t, a_t), r_t$      // Observe reward $r_t$ and new state $s_{t+1}$
        $\mathcal{D} \leftarrow \mathcal{D} \cup \{(s_t, a_t, r_t, s_{t+1})\}$      // Store the transition in the replay buffer
    **end for**
    **for** each gradient step **do**
        $\theta \leftarrow \theta - \lambda \nabla_\theta J_Q(\theta)$      // Update the critic parameters
        **meta-train:**
        Sample mini-batch $d_{trn}$ from $\mathcal{D}$
        $L^{\text{main}} \leftarrow Eqs.\ (1),\ (8)\ or\ (9)$      // Main actor loss
        $L_\omega^{\text{aux}} \leftarrow Eqs.\ (6)\ or\ (7)$      // Auxiliary actor loss from meta-critic
        $\phi_{old} = \phi - \alpha \nabla_\phi L^{\text{main}}$      // Update actor according to vanilla critic only
        $\phi_{new} = \phi_{old} - \alpha \nabla_\phi L_\omega^{\text{aux}}$      // Update actor according to meta-critic
        **meta-test:**
        Sample mini-batch $d_{val}$ from $\mathcal{D}$
        $L^{\text{meta}}(d_{val}; \phi_{old}, \phi_{new}) \leftarrow Eq.\ (5)$      // Meta-loss: Did meta-critic improve performance?
        **meta-optimization**
        $\phi \leftarrow \phi - \eta(\nabla_\phi L^{\text{main}} + \nabla_\phi L_\omega^{\text{aux}})$      // Update actor parameters
        $\omega \leftarrow \omega - \eta \nabla_\omega L^{\text{meta}}$      // Update meta-critic parameters
    **end for**
**end for**=0

---

samples, *after being trained by the meta-critic*. Note that in principle the lower-level optimization could purely rely on $L_\omega^{\text{aux}}$ analogously to the procedure in EPG (Houthooft et al., 2018), but we find that optimizing their linear combination greatly increases learning stability and speed. Eq. (3) is satisfied when the meta-critic successfully improves the actor's performance on the main task as measured by meta-loss. Note that the vanilla critic update is also in the lower loop, but as it updates as usual, so we focus on the actor and meta-critic optimization for simplicity of exposition.

In this setup the meta-critic is a neural network $h_\omega(d_{trn}; \phi)$ that takes as input some featurisation of the actor $\phi$ and the states and actions in $d_{trn}$. This auxiliary neural network must produce a scalar output, which we can then treat as a loss $L_\omega^{\text{aux}} := h_\omega$, and must be differentiable with respect to $\phi$. We next discuss the overall optimization flow, and discuss the specific meta-critic architecture later.

**Meta-Optimization Flow.** To optimize Eq. (3), we iteratively update the meta-critic parameters $\omega$ (upper-level) and actor and vanilla-critic parameters $\phi$ and $\theta$ (lower-level). At each iteration, we perform: (i) Meta-train: Sample a mini-batch of transitions and putatively update policy $\phi$ according to the main $L^{\text{main}}$ and meta-critic $L_\omega^{\text{aux}}$ losses. (ii) Meta-test: Sample another mini-batch of transitions to evaluate the performance of the updated policy according to $L^{\text{meta}}$. (iii) Meta-optimization: Update the meta-critic parameters $\omega$ to maximize the performance on the validation batch, and perform the real actor update according to both losses. In this way the meta-critic is trained online and in parallel to the actor so that they co-evolve. Figure 1 and Algorithm 1 summarize the process and the details of each step are explained next.

**Updating Actor Parameters ($\phi$).** During meta-train, we randomly sample a mini-batch of transitions $d_{trn} = \{(s_i, a_i, r_i, s_{i+1})\}$ with batch size $N$ from the replay buffer $\mathcal{D}$. We then update the pol-

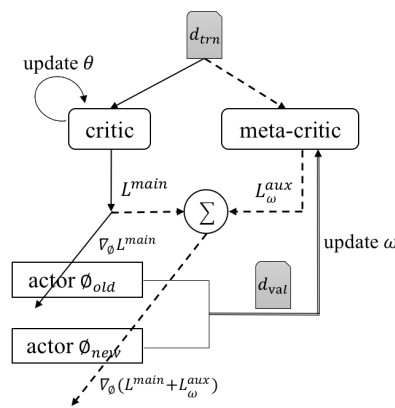

Figure 1: Meta-critic for Off-PAC. The agent uses data sampled from the replay buffer during meta-train and meta-test. Actor parameters are first updated using only vanilla critic, or both vanilla- and meta-critic. Meta-critic parameters are updated by the meta-loss.

icy using both losses as: $\phi_{new} = \phi - \eta \frac{\partial L^{\text{main}}(d_{trn})}{\partial \phi} - \eta \frac{\partial L_\omega^{\text{aux}}(d_{trn})}{\partial \phi}$. We also compute a separate update $\phi_{old} = \phi - \eta \frac{\partial L^{\text{main}}(d_{trn})}{\partial \phi}$ that only makes use of the vanilla loss. If the meta-critic provided a beneficial source of loss, $\phi_{new}$ should be a better parameter than $\phi$, and in particular it should be a better parameter than $\phi_{old}$. We will use this comparison in the next meta-test step.

**Updating Meta-Critic Parameters ($\omega$).** To train the meta-critic network, we sample another mini-batch of transitions: $d_{val} = \{(s_i^{\text{val}}, a_i^{\text{val}}, r_i^{\text{val}}, s_{i+1}^{\text{val}})\}$ with batch size $M$. The use of a validation batch for bi-level meta-optimization (Franceschi et al., 2018; Rajeswaran et al., 2019) ensures the meta-learned component does not overfit. Since our framework is off-policy, this does not incur any sample-efficiency cost. The meta-critic is then updated by a meta loss $\omega \leftarrow \underset{\omega}{\arg\min}\, L^{\text{meta}}(d_{val}; \phi_{new})$, which could in principle be the same as the main loss $L^{\text{meta}} = L^{\text{main}}$. However, we find it helpful for optimization efficiency to optimize the (monotonically related) difference between the updates with- and without meta-critic's input. Specifically, we use

$$L^{\text{meta}} = \tanh(L^{\text{main}}(d_{val}; \phi_{new}) - L^{\text{main}}(d_{val}; \phi_{old})), \tag{4}$$

which is simply a re-centering and re-scaling of $L^{\text{main}}$. This leads to

$$\omega \leftarrow \underset{\omega}{\arg\min}\, \tanh(L^{\text{main}}(d_{val}; \phi_{new}) - L^{\text{main}}(d_{val}; \phi_{old})). \tag{5}$$

Note that here the updated actor $\phi_{new}$ has dependence on the feedback given by meta-critic $\omega$ and $\phi_{old}$ does not. Thus only the first term is optimized for $\omega$. In his setup the $L^{\text{main}}(d_{val}; \phi_{new})$ term should obtain high reward/low loss on the validation batch and the latter provides a *baseline*, analogous to the baseline commonly used to accelerate and stabilize policy-gradient RL. The use of $\tanh$ reflects the idea of diminishing marginal utility, and ensures that the meta-loss range is always nicely distributed in $[-1, 1]$. In essence, the meta-loss is for the agent to ask itself the question based on the validation batch, "Did meta-critic improve the performance?", and adjusts the parameters of meta-critic accordingly.

**Designing Meta-Critic ($h_\omega$).** The meta-critic network $h_\omega$ implements the auxiliary loss for the actor. The design-space for $h_\omega$ has several requirements: (i) Its input must depend on the policy parameters $\phi$, because this auxiliary loss is also used to update policy network. (ii) It should be permutation invariant to transitions in $d_{trn}$, i.e., it should not make a difference if we feed the randomly sampled transitions indexed [1,2,3] or [3,2,1]. The most naive way to achieve (i) is given in MetaReg (Balaji et al., 2018) which meta-learns a parameter regularizer: $h_\omega(\phi) = \sum_i \omega_i|\phi_i|$. Although this form of $h_\omega$ acts directly on $\phi$, it does not exploit state information, and introduces a large number of parameters as $\phi$, and then $h_\omega$ may be a high-dimensional neural network. Therefore, we design a more efficient and effective form of $h_\omega$ that also meets both of these requirements. Similar to the feature extractor in supervised learning, the actor needs to analyse and extract information from states for decision-making. We assume the policy network can be represented as $\pi_\phi(s) = \hat{\pi}(\bar{\pi}(s))$ and decomposed into the feature extraction $\bar{\pi}_\phi$ and decision-making $\hat{\pi}_\phi$ (i.e., the last layer of the full policy network) modules. Thus the output of the penultimate layer of full policy network is just the output of feature extraction $\bar{\pi}_\phi(s)$, and such output of feature jointly encodes $\phi$ and $s$. Given this encoding, we implement $h_w(d_{trn}; \phi)$ as a three-layer multi-layer perceptron (MLP) whose input is the extracted feature from $\bar{\pi}_\phi(s)$. Here we consider two designs for meta-critic ($h_\omega$): using our joint feature alone (Eq. (6)) or augmenting the joint feature with states and actions (Eq. (7)):

$$(i) \qquad h_w(d_{trn}; \phi) = \frac{1}{N} \sum_{i=1}^{N} \text{MLP}_\omega(\bar{\pi}_\phi(s_i)), \tag{6}$$

$$(ii) \qquad h_w(d_{trn}; \phi) = \frac{1}{N} \sum_{i=1}^{N} \text{MLP}_\omega(\bar{\pi}_\phi(s_i), s_i, a_i). \tag{7}$$

$h_\omega$ is to work out the auxiliary loss based on such batch-wise set-embedding (Zaheer et al., 2017) of our joint actor-state feature. That is to say, $d_{trn}$ is a randomly sampled mini-batch transitions from the replay buffer, and then the $s$ (and $a$) of the transitions are inputted to the $h_\omega$ network in a permutation invariant way, and finally we can obtain the auxiliary loss for this batch $d_{trn}$. Here, our design of Eq. (7) also includes the cues features in LIRPG and EPG where $s_i$ and $a_i$ are used as the input of their learned reward and loss respectively. We set a softplus activation to the final layer of $h_\omega$, following the idea in TD3 that the vanilla critic may over-estimate and so the introduction of a non-negative actor auxiliary loss can mitigate such over-estimation. Moreover, we point out that

only $s_i$ (and $a_i$) from $d_{trn}$ are used when calculating $L^{\text{main}}$ and $L^{\text{aux}}_\omega$ for the actor, while $s_i$, $a_i$, $r_i$ and $s_{i+1}$ are all used for optimizing the vanilla critic.

**Implementation on DDPG, TD3 and SAC.** Our meta-critic module can be incorporated in the main Off-PAC methods DDPG, TD3 and SAC. In our framework, these algorithms differ only in their definitions of $L^{\text{main}}$, and the meta-critic implementation is otherwise exactly the same for each. Further implementation details can be found in the supplementary material.

TD3 (Fujimoto et al., 2018) borrows the Double Q-learning idea and use the minimum value between both critics to make unbiased value estimations. At the same time, computational cost is obtained by using a single actor optimized with respect to $Q_{\theta_1}$. Thus the corresponding $L^{\text{main}}$ for actor becomes:

$$L^{\text{main}} = -\mathbb{E}_{s \sim p_\pi} Q_{\theta_1}(s,a)|_{a=\pi_\phi(s)}. \tag{8}$$

In SAC, two key ingredients are considered for the actor: maximizing the policy entropy and automatic temperature hyper-parameter regulation. At the same time, the latest version of SAC (Haarnoja et al., 2018b) also draws lessons from "taking the minimum value between both critics". The $L^{\text{main}}$ for SAC actor is:

$$L^{\text{main}} = \mathbb{E}_{s \sim p_\pi}[\alpha \log\left(\pi_\phi(a|s)\right) - Q_\theta(s,a)|_{a=\pi_\phi(s)}]. \tag{9}$$

# 4 EXPERIMENTS AND EVALUATION

The goal of our experimental evaluation is to demonstrate the versatility of our meta-critic module in integration with several prior Off-PAC algorithms, and its efficacy in improving their respective performance. We use the open-source implementations of DDPG, TD3 and SAC algorithms as our baselines, and denote their enhancements by meta-critic as DDPG-MC, TD3-MC, SAC-MC respectively. All -MC agents have both their built-in vanilla critic, and the meta-critic that we propose. We take Eq. (6) as the default meta-critic architecture $h_\omega$, and we compare the alternative in the later ablation study. For our implementation of meta-critic, we use a three-layer neural network with an input dimension of $\bar{\pi}$ (300 in DDPG and TD3, 256 in SAC), two hidden feed-forward layers of 100 hidden nodes each, and ReLU non-linearity between layers.

We evaluate the methods on a suite of seven MuJoCo continuous control tasks (Todorov et al., 2012) interfaced through OpenAI Gym (Brockman et al., 2016) and HalfCheetah and Ant (Duan et al., 2016a) in rllab. We use the latest V2 tasks instead of V1 used in TD3 and the old implementation of SAC (Haarnoja et al., 2018a) without any modification to their original environment or reward.

**Implementation Details.** For DDPG, we use the open-source implementation "OurDDPG" [1] which is the re-tuned version of DDPG implemented in Fujimoto et al. (2018) with the same hyper-parameters of the actor and critic. For TD3 and SAC, we use the open-source implementations of TD3 [2] and SAC [3]. In each case we integrate our meta-critic with learning rate 0.001. The specific pseudo-codes can be found in the supplementary material.

## 4.1 EVALUATION OF META-CRITIC OFF-PAC LEARNING

**DDPG** Figure 2 shows the learning curves of DDPG and DDPG-MC. The experimental results corresponding to each task are averaged over 5 random seeds (trials) and network initialisations, and the standard deviation confidence intervals are represented as shaded regions over the time steps. Following Fujimoto et al. (2018), curves are uniformly smoothed (window size 30) for clarity. We run the gym-MuJoCo experiments for 1-10 million depen ding on to environment, and rllab experiments for 3 million steps. Every 1000 steps we evaluate our policy over 10 episodes with no exploration noise.

From the learning curves in Figure 2, we can see that DDPG-MC generally outperforms the corresponding DDPG baseline in terms of the learning speed and asymptotic performance. Furthermore, it usually has smaller variance. The summary results for all nine tasks in terms of max average return are given in Table 1. We selected the six tasks shown in Figure 2 for plotting, because the

---

[1] https://github.com/sfujim/TD3/blob/master/OurDDPG.py
[2] https://github.com/sfujim/TD3/blob/master/TD3.py
[3] https://github.com/pranz24/pytorch-soft-actor-critic

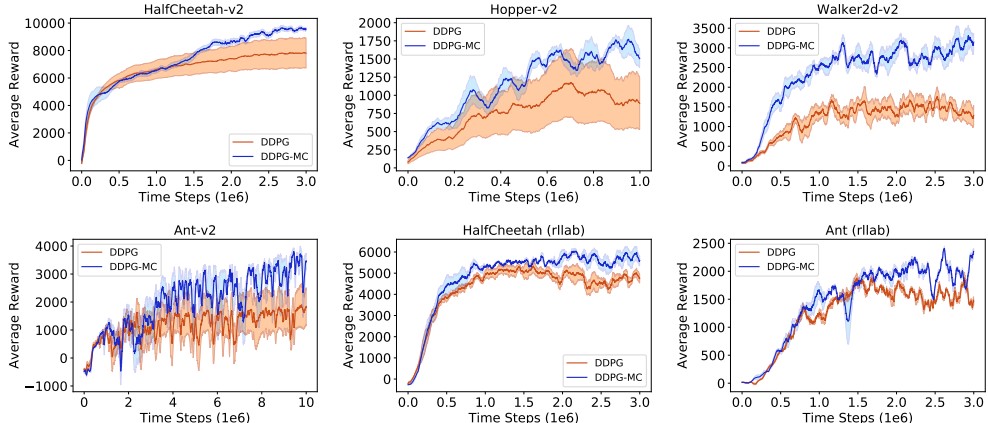

Figure 2: Learning curve mean and standard-deviation of vanilla DDPG and meta-critic enhanced DDPG-MC for continuous control tasks.

Table 1: Max Average Return over 5 trials over all time steps. Max value for each task is bolded.

| Environment | DDPG | DDPG-MC | TD3 | TD3-MC | SAC | SAC-MC | PPO | PPO-LIRPG |
|---|---|---|---|---|---|---|---|---|
| HalfCheetah | 8440.2 | 10187.5 | 12735.7 | 15064.0 | 16651.8 | **16815.9** | 2061.5 | 1882.6 |
| Hopper | 2097.5 | 3253.6 | 3807.0 | **3854.3** | 3610.6 | 3738.4 | 3762.0 | 2750.0 |
| Walker2d | 2920.1 | 3753.7 | 5942.7 | 5955.5 | 6398.8 | **7164.9** | 4432.6 | 3652.9 |
| Ant | 2375.4 | 3661.1 | 5914.8 | 6280.0 | 6954.4 | **7204.3** | 684.2 | 23.6 |
| Reacher | -3.6 | -3.7 | -3.0 | -2.9 | -2.8 | **-2.7** | -6.08 | -7.53 |
| InvPend | **1000.0** | **1000.0** | **1000.0** | **1000.0** | **1000.0** | **1000.0** | 988.2 | 971.6 |
| InvDouPend | 9307.5 | 9326.5 | 9357.4 | 9358.8 | **9359.6** | **9359.6** | 7266.0 | 6974.9 |
| HalfCheetah(rllab) | 5860.8 | 6254.6 | 8029.6 | 8552.1 | 10011.0 | **10597.0** | - | - |
| Ant(rllab) | 2300.8 | 2721.1 | 3672.6 | 4776.8 | 8014.8 | **8353.8** | - | - |

other MuJoCo tasks "Reacher", "InvertedPendulum" and "InvertedDoublePendulum" have an environmental reward upper bound which all methods reach quickly without obvious difference between them. Table 1 shows that DDPG-MC provides consistently higher max return for the tasks without upper bounds.

**TD3 and SAC** Figure 3 reports the learning curves for TD3. For some tasks vanilla TD3 performance declines in the long run, while our TD3-MC shows improved stability with much higher asymptotic performance. Generally speaking, the learning curves show that TD3-MC providing comparable or better learning performance in each case, while Table 1 shows the clear improvement in the max average return. Figure 4 report the learning curves of SAC. Note that we use the most recent update of SAC (Haarnoja et al., 2018b), which can be regarded as the combination SAC+TD3. Although this SAC+TD3 is arguably the strongest existing method, SAC-MC still gives a clear boost on the asymptotic performance for several of the tasks.

**Comparison vs PPO-LIRPG** Intrinsic Reward Learning for PPO (Zheng et al., 2018) is the most related method to our work in performing online single-task meta-learning of an auxiliary reward/loss via a neural network. The original PPO-LIRPG study evaluated on a modified environment with hidden rewards. Here we apply it to the standard unmodified learning tasks that we aim to improve. The results in Table 1 demonstrate that: (i) In this conventional setting, PPO-LIRPG worsens rather than improves basic PPO performance. (ii) Overall Off-PAC methods generally perform better than on-policy PPO for most environments. This shows the importance of our meta-learning contribution to the off-policy setting. In general Meta-Critic is preferred compared to PPO-LIRPG because the latter only provides a scalar reward bonus only influences the policy indirectly via policy-gradient updates, while Meta-Critic provides a direct loss.

**Summary** Table 1 and Figure 5 summarize all the results in terms of max average return. We can see that SAC-MC always performs best; the Meta-Critic-enhanced methods are generally comparable

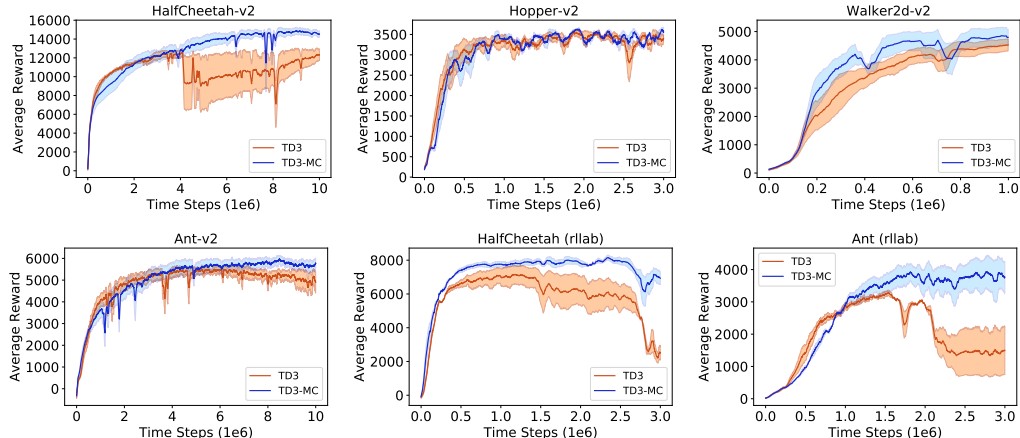

Figure 3: Learning curve mean and standard-deviation of vanilla TD3 and meta-critic enhanced TD3-MC for continuous control tasks.

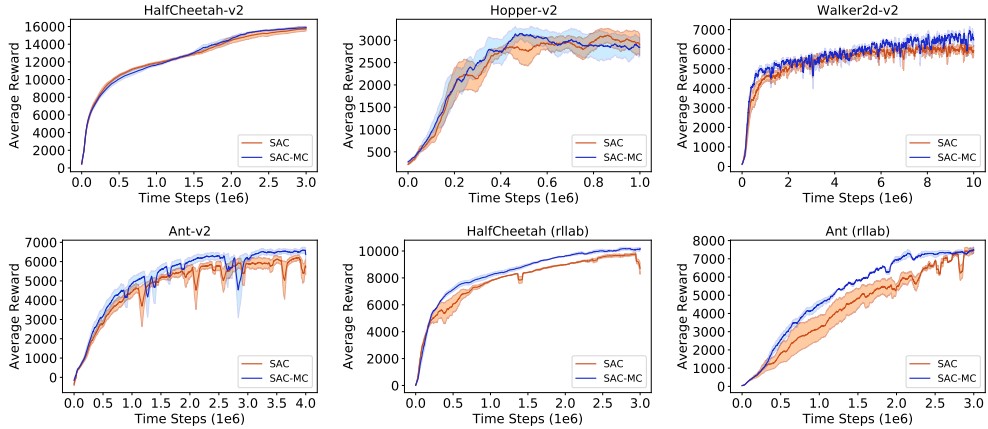

Figure 4: Learning curve mean and standard-deviation of vanilla SAC and meta-critic enhanced SAC-MC for continuous control tasks.

or better than their corresponding vanilla alternatives; and Meta-Critic usually provides improved variance in return compared to the baselines.

## 4.2 FURTHER ANALYSIS

**Loss Analysis.** To analyse the learning dynamics of our algorithm, we take Walker2d as an example. Figure 6 reports the main loss $L^{\text{main}}$ curve of actor and the loss curves of $h_\omega$ (i.e., $L_\omega^{\text{aux}}$) and $L^{\text{meta}}$ over 5 trials for SAC. We can see that: (i) SAC-MC shows faster convergence to a lower value of $L^{\text{main}}$, demonstrating the auxiliary loss's ability to accelerate learning. Unlike supervised learning, where the vanilla loss is, e.g., cross-entropy vs ground-truth labels. The $L^{\text{main}}$ for actors in RL is provided by the critic which is also learned, so the plot also encompasses convergence of the critic. (ii) The meta-loss (which corresponds to the success of the meta-critic in improving actor learning) fluctuates throughout, reflecting the exploration process in RL. But it is generally negative, confirming that the auxiliary-trained actor generally improves on the vanilla actor at each iteration. (iii) The auxiliary loss converges smoothly under the supervision of the meta-loss.

**Ablation on $h_\omega$ design.** We also run Walker2d experiments with alternative $h_\omega$ designs as in Eq. (7) or MetaReg (Balaji et al., 2018) format (input actor parameters directly). As shown in Table 2, we record the max average return and sum average return (regarded as the area under the average reward curve) of all evaluations during all time steps. Eq. (7) achieves the highest max average return and

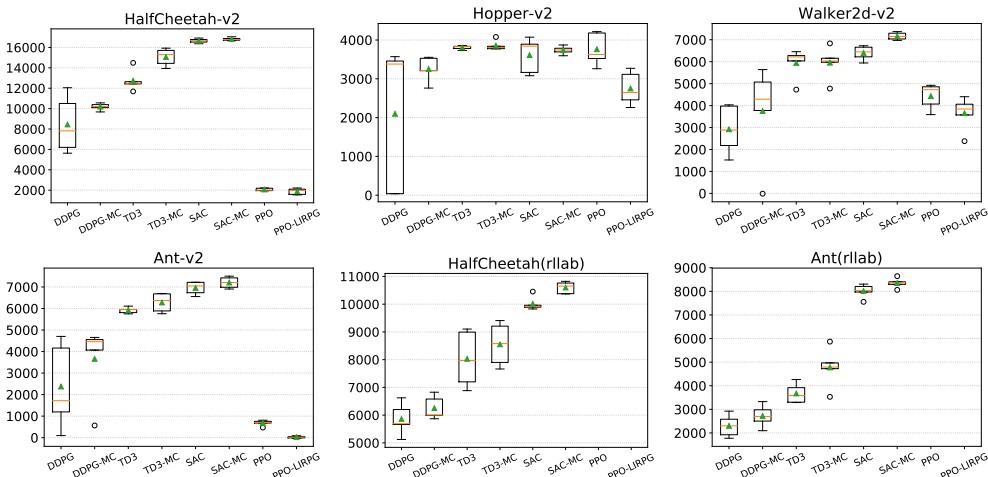

Figure 5: Box plots of the Max Average Return over 5 trials of all time steps.

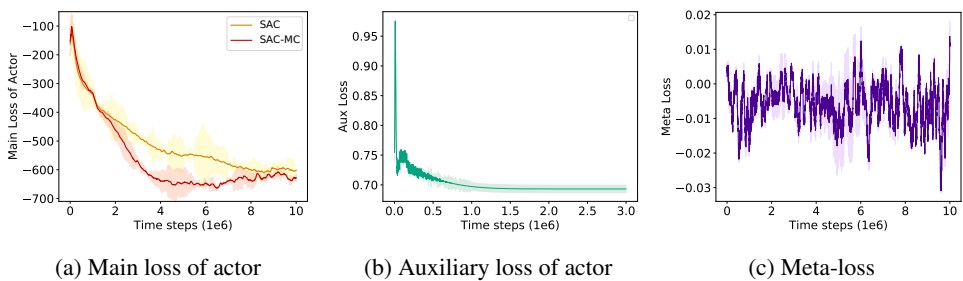

(a) Main loss of actor    (b) Auxiliary loss of actor    (c) Meta-loss

Figure 6: Loss analysis of our algorithm.

our default $h_\omega$ (Eq. (6)) attains the highest mean average return. We can also see some improvement for $h_\omega(\phi)$ using MetaReg format, but the huge number (73484) of parameters is expensive. Overall, all meta-critic module designs provides at least a small improvement on vanilla SAC.

**Ablation on baseline in meta-loss.** In Eq. (5), we use $L^{\mathrm{main}}(d_{val}; \phi_{old})$ as a baseline to improve numerical stability of the gradient update. To evaluate this design, we remove the $\phi_{old}$ baseline and optimize $\omega \leftarrow \underset{\omega}{\operatorname{argmin}} \tanh(L^{\mathrm{main}}(d_{val}; \phi_{new}))$. The last column in Table 2 shows that this barely improves on vanilla SAC, validating our design choice to use a baseline.

Table 2: Max and Sum Average Return over 5 trials of all time steps under different designs of meta-critic (aux-loss) and meta-loss. Max value in each row is bolded.

| | SAC | $L^{\mathrm{meta}} : \phi_{new} - \phi_{old}$ | | | $L^{\mathrm{meta}} : \phi_{new}$ |
| --- | --- | --- | --- | --- | --- |
| | | $h_\omega(\bar{\pi}_\phi)$ | $h_\omega(\bar{\pi}_\phi, s, a)$ | $h_\omega(\phi)$ | $h_\omega(\bar{\pi}_\phi)$ |
| Max Average Return | $6398.8 \pm 289.2$ | $7164.9 \pm 151.3$ | $\mathbf{7423.8 \pm 780.2}$ | $6644.3 \pm 1815.6$ | $6456.1 \pm 424.8$ |
| Sum Average Return | 53,695,678 | **61,672,039** | 57,364,405 | 58,875,184 | 52,446,717 |

## 5 CONCLUSION

We present Meta-Critic, an auxiliary critic module for Off-PAC methods that can be meta-learned online during single task learning. The meta-critic is trained to generate gradients that improve the actor's learning performance over time, and leads to long run performance gains in continuous control. The meta-critic module can be flexibly incorporated into various contemporary Off-PAC methods to boost performance. In future work, we plan to apply the meta-critic to conventional offline meta-learning with multi-task and multi-domain RL.

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

# Supplementary Information

## 6 ALGORITHMS OF META-CRITIC FOR DDPG, TD3 AND SAC

We incorporate our Meta-Critic to the implementation of vanilla DDPG, TD3 and SAC, following their original implementations.

---

**Algorithm 2** DDPG-MC algorithm

---

Initialize critic $Q(s, a|\theta)$, actor $\pi(s|\phi)$ and auxiliary loss network $h_\omega$
Initialize target network $Q'$ and $\pi'$ with weights $\theta' \leftarrow \theta$, $\phi' \leftarrow \phi$
Initialize replay buffer $\mathcal{R}$
**for** episode = 1, ..., M **do**
    Initialize a random process $\mathcal{N}$ for action exploration
    Receive initial observation state $s_1$
    **for** t = 1, ..., T **do**
        Select action $a_t = \pi(s_t|\phi) + \mathcal{N}_t$ according to the current policy and exploration noise
        Execute action $a_t$, observe reward $r_t$ and new state $s_{t+1}$
        Store transition $(s_t, a_t, r_t, s_{t+1})$ in $\mathcal{R}$
        Sample a random mini-batch of $N$ transitions $(s_i, a_i, r_i, s_{i+1})$ from $\mathcal{R}$
        Set $y_i = r_i + \gamma Q'(s_{i+1}, \pi'(s_{i+1}|\phi')|\theta')$
        Update critic by minimizing the loss: $L = N^{-1} \sum_i (y_i - Q(s_i, a_i|\theta))^2$
        **meta-train**:
        Calculate the old actor weights using the main actor loss:

$$\nabla_\phi L^{\text{main}} = -N^{-1} \sum_i \nabla_a Q(s, a|\theta)|_{s=s_i, a=\pi(s)} \nabla_\phi \pi(s|\phi)|_{s=s_i}$$

$$\phi_{old} = \phi - \alpha \nabla_\phi L^{\text{main}}$$

        Calculate the new actor weights using the auxiliary actor loss:

$$\nabla_\phi L_\omega^{\text{aux}} = \nabla_\phi h_\omega = N^{-1} \sum_i \nabla_\phi MLP_\omega(\bar{\pi}(s|\phi)|_{s=s_i})$$

$$\phi_{new} = \phi_{old} - \alpha \nabla_\phi L_\omega^{\text{aux}}$$

      **meta-test**:
      Sample a random mini-batch of $N$ $s_i^{\text{val}}$ from $\mathcal{R}$
      Calculate the meta-loss using the meta-test sampled transitions:

$$L^{\text{meta}} = \tanh(L^{\text{main}}(s, a|\theta)|_{s=s_i^{\text{val}}, a=\pi(s|\phi_{new})} - L^{\text{main}}(s, a|\theta)|_{s=s_i^{\text{val}}, a=\pi(s|\phi_{old})})$$

      **meta-optimization**: Update the weight of actor and meta-critic network:

$$\phi \leftarrow \phi - \eta(\nabla_\phi L^{\text{main}} + \nabla_\phi L_\omega^{\text{aux}})$$

$$\omega \leftarrow \omega - \eta \nabla_\omega L^{\text{meta}}$$

      Update the target networks:

$$\theta' \leftarrow \tau\theta + (1 - \tau)\theta'$$

$$\phi' \leftarrow \tau\phi + (1 - \tau)\phi'$$

    **end for**
  **end for**=0

---

---

**Algorithm 3** TD3-MC algorithm

---

Initialize critics $Q_{\theta_1}$, $Q_{\theta_2}$, actor $\pi_\phi$ and auxiliary loss network $h_\omega$
Initialize target networks $\theta_1' \leftarrow \theta_1$, $\theta_2' \leftarrow \theta_2$, $\phi' \leftarrow \phi$
Initialize replay buffer $\mathcal{B}$
**for** $t = 1$ **to** $T$ **do**
    Select action with exploration noise $a \sim \pi_\phi(s) + \epsilon$, $\epsilon \sim \mathcal{N}(0, \sigma)$ and observe reward $r$ and new state $s'$
    Store transition tuple $(s, a, r, s')$ in $\mathcal{B}$

    Sample mini-batch of $N$ transitions $(s, a, r, s')$ from $\mathcal{B}$
    $\tilde{a} \leftarrow \pi_{\phi'}(s') + \epsilon$,    $\epsilon \sim clip(\mathcal{N}(0, \tilde{\sigma}), -c, c)$
    $y \leftarrow r + \gamma \min_{i=1,2} Q_{\theta_i'}(s', \tilde{a})$
    Update critics $\theta_i \leftarrow \arg\min_{\theta_i} N^{-1} \sum (y - Q_{\theta_i}(s, a))^2$
    **if** $t \bmod d$ **then**
        $\nabla_\phi L^{\mathrm{main}} = -N^{-1} \sum \nabla_a Q_{\theta_1}(s, a)|_{a=\pi_\phi(s)} \nabla_\phi \pi_\phi(s)$
        $\nabla_\phi L_\omega^{\mathrm{aux}} = \nabla_\phi h_\omega = N^{-1} \sum \nabla_\phi MLP_\omega(\bar{\pi}_\phi(s))$
        **meta-train** :
        Calculate the old actor weights using the main actor loss: $\phi_{old} = \phi - \alpha \nabla_\phi L^{\mathrm{main}}$
        Calculate the new actor weights using the auxiliary actor loss: $\phi_{new} = \phi_{old} - \alpha \nabla_\phi L_\omega^{\mathrm{aux}}$
        **meta-test**:
        Sample mini-batch of $N$ $s^{\mathrm{val}}$ from $\mathcal{B}$
        Calculate the meta-loss using the meta-test sampled transitions:
        $L^{\mathrm{meta}} = \tanh(L^{\mathrm{main}}(s^{\mathrm{val}}, a|\theta_1)|_{a=\pi(s^{\mathrm{val}})|\phi_{new}} - L^{\mathrm{main}}(s^{\mathrm{val}}, a|\theta_1)|_{a=\pi(s^{\mathrm{val}})|\phi_{old}})$
        **meta-optimization**:
        Update the actor and meta-critic:
        $\phi \leftarrow \phi - \eta(\nabla_\phi L^{\mathrm{main}} + \nabla_\phi L_\omega^{\mathrm{aux}})$
        $\omega \leftarrow \omega - \eta \nabla_\omega L^{\mathrm{meta}}$

        Update target networks:
        $\theta_i' \leftarrow \tau\theta_i + (1-\tau)\theta_i'$
        $\phi' \leftarrow \tau\phi + (1-\tau)\phi'$
    **end if**
**end for**=0

---

---

**Algorithm 4** SAC-MC algorithm

---

$\theta_1, \theta_2, \phi, \omega$      // Initialize parameters
$\bar{\theta} \leftarrow \theta_1, \bar{\theta}_2 \leftarrow \theta_2$      // Initialize target network weights
$\mathcal{D} \leftarrow \emptyset$      // Initialize an empty replay pool
**for** each iteration **do**
    **for** each environment step **do**
        $a_t \sim \pi_\phi(a_t|s_t)$      // Sample action from the policy
        $s_{t+1} \sim p(s_{t+1}|s_t, a_t)$      // Sample transition from the environment
        $\mathcal{D} \leftarrow \mathcal{D} \cup \{(s_t, a_t, r(s_t, a_t), s_{t+1})\}$      // Store the transition in the replay pool
    **end for**
    **for** each gradient step **do**
        $\theta_i \leftarrow \theta_i - \lambda_Q \nabla_{\theta i} J_Q(\theta_i)$ for $i \in \{1, 2\}$ // Update the Q-function parameters
        **meta-train** :
        $\nabla_\phi L^{\text{main}} = N^{-1} \sum_t \nabla_a [\alpha \log(\pi_\phi(a|s)) - Q_\theta(s, a)|_{s=s_t, a=\pi(s)}] \nabla_\phi \pi_\phi(s)|_{s=s_t}$
        $\phi_{old} = \phi - \alpha \nabla_\phi L^{\text{main}}$      // Calculate old weights of the actor
        $\nabla_\phi L_\omega^{\text{aux}} = \nabla_\phi h_\omega = N^{-1} \sum_t \nabla_\phi MLP_\omega(\bar{\pi}_\phi(s))|_{s=s_t}$
        $\phi_{new} = \phi_{old} - \alpha \nabla_\phi L_\omega^{\text{aux}}$      // Calculate new weights of the actor
        **meta-test**:
        $L^{\text{meta}} = \tanh(L^{\text{main}}(s, a|\theta)|_{s=s_t^{\text{val}}, a=\pi(s|\phi_{new})} - L^{\text{main}}(s, a|\theta)|_{s=s_t^{\text{val}}, a=\pi(s|\phi_{old})})$
                 // Calculate meta-loss
        **meta-optimization**:
        $\phi \leftarrow \phi - \eta(\nabla_\phi L^{\text{main}} + \nabla_\phi L_\omega^{\text{aux}})$      // Update the actor parameters
        $\omega \leftarrow \omega - \eta \nabla_\omega L^{\text{meta}}$      // Update the meta-critic parameters

        $\alpha \leftarrow \alpha - \lambda \nabla_\alpha J(\alpha)$      // Adjust temperature
        $\bar{\theta}_i \leftarrow \tau \theta_i + (1 - \tau)\bar{\theta}_i$ for $i \in \{1, 2\}$      // Update target network weights
    **end for**
**end for** =0

---

# 7 AVERAGE REWARDS ON OTHER TASKS AND PPO-LIRPG EXPERIMENTS

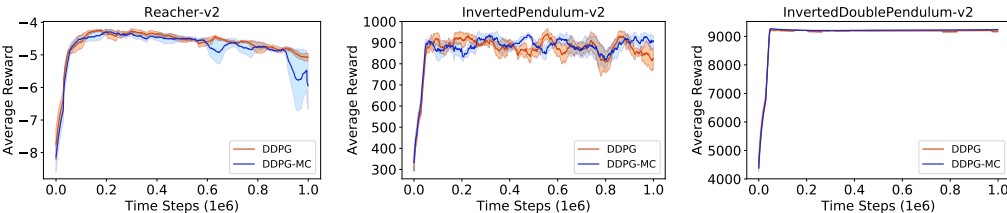

Figure 7: Learning curve mean and standard-deviation of vanilla DDPG and meta-critic enhanced DDPG-MC for MuJoCo tasks with upper reward bound.

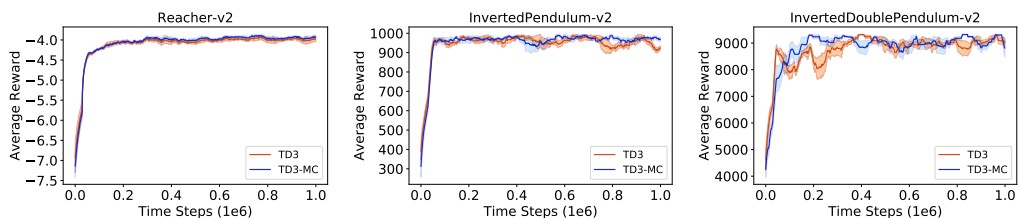

Figure 8: Learning curve mean and standard-deviation of vanilla TD3 and meta-critic enhanced TD3-MC for MuJoCo tasks with upper reward bound.

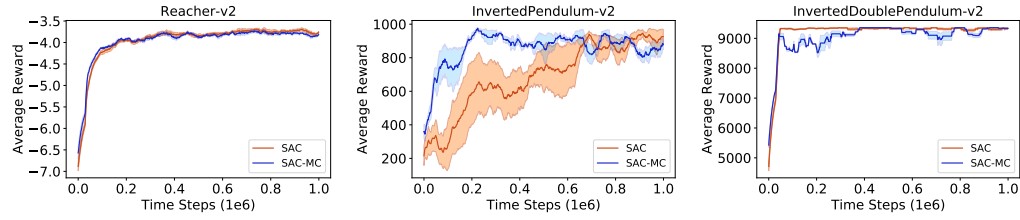

Figure 9: Learning curve mean and standard-deviation of vanilla SAC and meta-critic enhanced SAC-MC for MuJoCo tasks with upper reward bound.

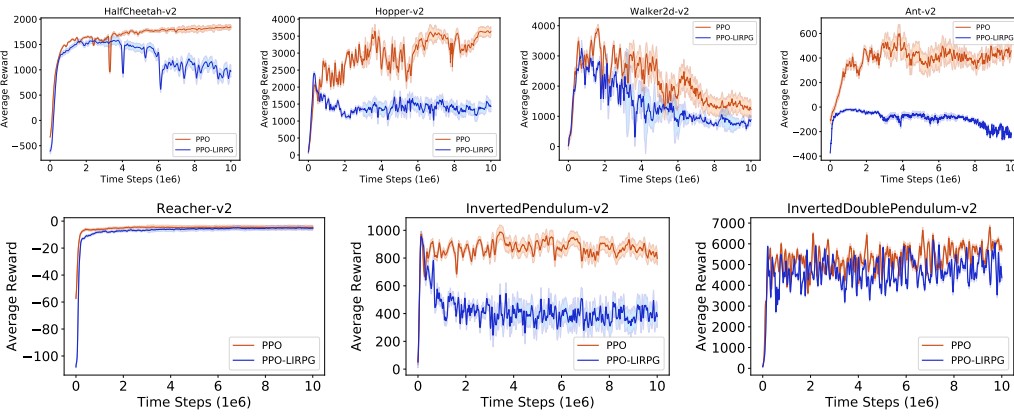

Figure 10: Learning curve mean and standard-deviation of PPO and PPO-LIRPG for continuous control tasks.

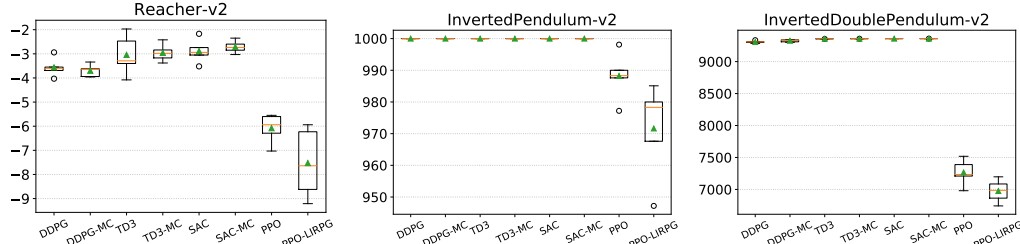

Figure 11: Box plots of the Max Average Return over 5 trials of all time steps for MuJoCo tasks with upper reward bound.

# 8 FURTHER ANALYSIS

## 8.1 META-CRITIC COMPUTATION

In terms of computation requirement, meta-critic takes around 15-30% more time per iteration, depending on the base algorithm. This is primarily attributable to the cost of evaluating the meta-loss $L^{\text{meta}}$, and hence $L^{\text{main}}$.

To investigate whether the benefit of meta-critic comes solely the additional compute expenditure, we perform an additional experiment where we increase the compute applied by the baselines to a corresponding degree. Specifically, if meta-critic takes $K\%$ more time than the baseline, then we re-run the baseline with $K\%$ more update steps iteration. This provides the baseline more mini-batch samples while controlling the number of environment interactions. Examples in Figure 12 shows that increasing the number of update steps does not have a straightforward link to performance. For DDPG, Walker2d-v2 performance increases with more steps, but stills performs worse than Meta-Critic. Meanwhile, for HalfCheetah, the extra iterations dramatically exacerbates the drop in performance that the baseline already experiences after around 1.5 million steps. Overall, there is no consistent impact of providing the baseline more iterations, and Meta-Critic's consistently good performance can not be simply replicated by a corresponding increase in gradient steps taken by the baseline.

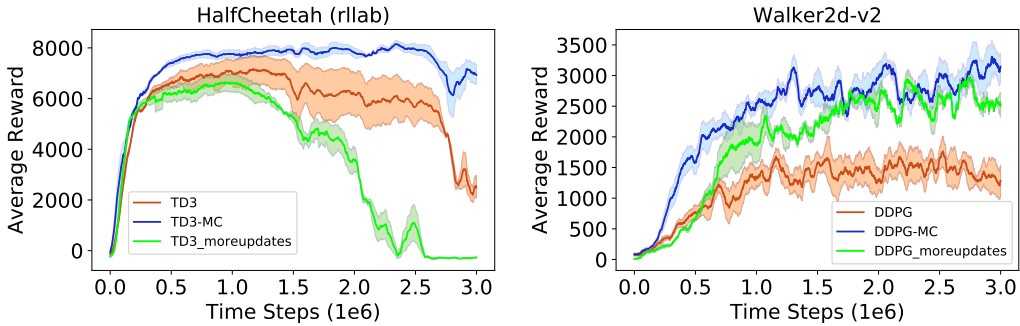

Figure 12: Experiment controlling for compute time per method. Assigning more update iterations to the baselines so their running speed matches Meta-Critic. Left: Learning curves of TD3 HalfCheetah (rllab). Right: Learning curves of DDPG Walker2d-v2.

## 8.2 ADDITIONAL ENVIRONMENTS

In order to investigate the impact of meta-critic on harder environments, we evaluated SAC and SAC-MC on TORCS and Humanoid(rllab). The results in Figure 13 show that meta-critic provides a clear margin of performance improvement in these more challenging environments.

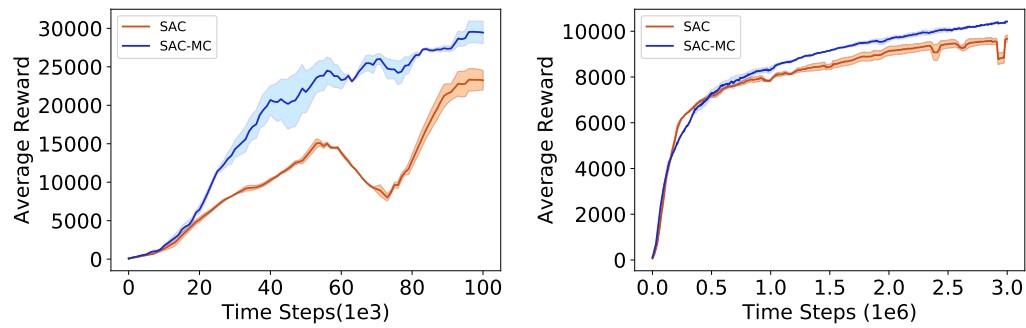

Figure 13: Learning curves of SAC and SAC-MC on the TORCS driving (Left) and Humanoid (Right) environments

.

