# OpenReview forum: "Online Meta-Critic Learning for Off-Policy Actor-Critic Methods"
_ICLR.cc/2020/Conference — Reject_

### Official Review · AnonReviewer3 · 2019-10-16
**Official Blind Review #3**

**Rating:** 3

**Review:**

This paper proposes a meta reinforcement learning approach where a meta-critic is trained in addition to a conventional actor and critic, so that the meta-critic boosts the training of the actor over a single task. The approach is combined to DDPG, TD3 and SAC and is claimed to convey superior performance (or learning speed) over these state-of-the-art actor-critic algorithms.

At first glance, the paper looks convincing, but closer inspection reveals a potential issue that I would like the authors to discuss.

The first cue is that in Fig. 6, the meta-loss does not seem to converge to anything, as the authors say it just fluctuates. Shouldn't it converge once the actor converges to close to optimal performance?

The second cue is that appart from the rllab tasks (ant and half-cheetah), it is not clear that the meta-critic approach brings some gain in the end of training. Particularly in Reacher (Fig 7), the performance seems to collapse faster with DDPG-MC than with DDPG, and on the rest of curves of Figs 7 and 8 it is had to determine whether the MC approach brings something significant or not. And in Fig. 3, in Walker2D, TD3-MC looks rather unstable.

The third cue is that in Section 3.2, the paragraphs "Updating MC parameters" and "Designing MC" are rather unclear (I'll come back to that) and lack a theoretical justification.

So I'm wondering what exactly MC is doing and I would like to see a more detailled analysis. Couldn't a similar performance improvement be obtained by just increasing the actor learning rate and addiing some noise to the actor learning gradient? Isn't this more or less what the meta-loss does, when looking at Fig. 6?

To me, unless the authors give a clear answer to the points above, the paper should be rejected as it does not provide clear enough evidence and justification in favor of the proposed meta-learning approach.

As stated above, I found the paragraph "Updating MC parameters" lacking a principled justification.

About "Designing MC", it could be much clearer. You first express two requirements (i) and (ii). Fine.
Then you explain how to meet requirement (i), but you say this is not what you are doing (!), and then you try to explain what you are doing but without going back to the requirements. In particular, it is not clear at all why your design is "permutation invariant". You should be much more direct. Besides, the way to extract features and the relationship to batch-wise set-embedding should be made more explicit. Well, it is complicated and should be explained more clearly...

"TD3 borrows the Double Q learning idea...": no, TD3 does more than this, as it uses the min between both critics, which double Q-learning and DDQN do not.

You use HalfCheetah-v2 from gym-mujoco and HalfCheetah from rllab. Why? One might suspect this is because the performance of the MC appraoch is only better with rllab...

You say you are using 10 million steps, but this is not always the case.

It is interesting to see that the performance of TD3 on Half-Cheetah collapses at around 4 million steps. Any idea why? It seems that in many papers experiments are stopped before performance collapses, and a strong study about this phenomenon is missing as the authors don't want to show this. This is an open call to readers: a paper on that would be great! :)

"SAC-MC" gives a clear-boost for several tasks": well, looking at the figures it is not so clear. Do you mean faster learning, higher final performance or both? I would like to see this claim backed-up by some proper statistical significance test and a clear specification of the number of seeds, etc. Fig.5 conveys the adequate data for such test if your claim is that this is the final performance that matters (but beware that curves are crossing eachother, so the final performance depends a lot on where you stop...).

typos:

p1 "For example, ... (Zheng et al., 2018)." is not a sentence (no main verb).
p2 "the on-policy (approach?) needs to interact with (the) environment..."
p2 "is less effective than off-policy transitions." => unclear statement
p3 to assist(s)

Equations (2), (6) and (8) should finish with a dot as they close a sentence.

Eq X => Eq (X) : use \eqref{label} instead of just \ref{label}

p5 It's input => Its
p5 two key technologies: I would not call this a technology. Ingredients?
p5 "computational cost is ? by using": missong word

p6: asmyptotic

p9: we removing


**Experience Assessment:**

I have read many papers in this area.

**Review Assessment: Checking Correctness Of Derivations And Theory:**

I did not assess the derivations or theory.

**Review Assessment: Checking Correctness Of Experiments:**

I assessed the sensibility of the experiments.

**Review Assessment: Thoroughness In Paper Reading:**

I read the paper at least twice and used my best judgement in assessing the paper.

---

> ### Author Response · Authors · 2019-11-13
> **Author response to Review #3 (part 1)**
>
> Thanks for your comments and questions. We address them below.
>
> Q1: Should the meta-loss converge?
>
> A1: No. The salient point is that the main and auxiliary losses do converge. Our meta loss per-se is *not* expected to converge. To understand why, recall that the meta-loss is implemented as $L^{meta} = tanh(L^{main}(d_{val};\phi_{new}) - L^{main}(d_{val};\phi_{old}))$. The consider:
>
> (i) The meta-loss reflects the *degree of improvement* by the auxiliary loss updated policy when compared to the conventionally updated policy. That *degree of improvement* obtained is something that naturally varies across iterations according to the exploration process in RL, etc. The key point is that it is fairly consistently *negative*, indicating successful improvement. By analogy, you can imagine interpreting the convergence of any RL algorithm not by its loss or return curve, but by its $Delta(t)=Loss(t)-Loss(t-1)$ curve with respect to $t$. Such a curve should be negative, but need not converge if there are fluctuations at each iteration.
>
> (ii) The only learn-able term in the meta-loss is the first term, $L^{main}(\phi_{new})$. The second term is treated as a constant. Therefore the meta loss is a re-centering (by $L^{main}(\phi_{old})$) and re-scaling (by $tanh$) of the main loss. The first, and only learn-able, term $L^{main}(\phi_{new})$ *does* converge, as shown in Fig 6(a). So the meta-loss does converge in the sense that its only learn-able component converges.
>
> (iii) Conceptually, we could optimize $L^{meta}=L^{main}$ directly without the re-scaling, and then Fig 6c would be exactly the same quantity as Fig 6a. But this design is shown not to work well in the ablation of Tab 2 (right column). Empirically we found that optimizing our dynamically recentered/rescaled meta-loss to perform better, because the loss values to optimize are always in a nice small range (Fig 6c).
>
> Q2: Consistency and stability of performance improvement: Performance at end of training for Reacher-DDPG. Fig 7 \& 8. Walker-TD3.
>
> A2-1: There are several ways to evaluate RL algorithms: Max return, performance at the end of training, and area under return curve. The last two are sensitive to stopping time, especially given the unfortunate tendency of existing RL algorithms to collapse from a good solution (solving this is out of the scope of our work). We focused on max-return in Tab 1 and Fig 5, where we see a consistent benefit. Our MC asymptotic return is indeed worse in the specific case of DDPG-Reacher-V2. But asymptotic return is arguably same or better for all the other cases of MC-enhanced runs compared to the vanilla algorithms. This should not be a major problem: Most published RL algorithms do not improve on previous competitors by every metric in every environment. Furthermore, while MC only slightly accelerated the collapse problem in DDPG-Reacher-V2, it significantly ameliorated it in many more environments including TD3-HalfChetah-v2, TD3-HalfCheetah-rllab, TD3-Ant-rllab.
>
> A2-2: Fig 7\&8 are the tasks where all algorithms saturate near the maximum possible reward for the environments. We include them in an appendix for completeness, and to confirm that MC algorithms are comparable to vanilla algorithms in these cases. But since they are essentially solved environments anyway, we do not expect improvements here.
>
> A2-3: Yes, Walker TD3-MC has minor instability. However, overall meta-critic is more stable (fewer big drops); and less seed sensitive (lower variance) than the baselines. See DDPG: All Fig 2 examples. TD3: Half Cheetah-v2, Half Cheetah-rllab, Ant-rllab. SAC: Ant rllab.  etc.
>
> Q3: What exactly is meta-critic doing?
>
> A3: Please see R1 Q3.

---

> ### Author Response · Authors · 2019-11-13
> **Author response to Review #3 (part 2)**
>
> Q4: Could a similar performance improvement be obtained by just increasing the actor learning rate and adding some noise to the actor learning gradient? Isn't this more or less what the meta-loss does, when looking at Fig. 6?
>
> A4: (i) Perhaps carefully designed noise or learning rates can help in RL. But these hyperparameters are incredibly challenging to tune, and furthermore would need to be done on an environment-specific basis. While meta-critic is a black-box, one possible mechanism for its influence is via automatic tuning of an effective learning rate (see also R1: A3-2), where this adapting learning rate module is meta-learned online. (ii) Nevertheless, Fig 6 does *not* show that MC acts via "faster learning rate+noise": The faster loss minimisation in Fig 6a is *not* due to a faster learning rate per-se, or achievable by a faster learning rate. (If fast reinforcement learning were simple a matter of increasing learning rate, RL would be easy.) In practice larger learning rates can cause divergence, so there is no free lunch. Instead Fig 6a is the result of the overall dynamics of meta-critic learning. Fig 6b is *not* "adding gradient noise". It shows the degree of improvement of meta-critic over vanilla updates at each iteration. That improvement is always there (so it‘s negative), but of variable size (so it’s noisy). See also R3Q1.
>
> Q5: Designing Meta Critic and requirements (i) and (ii). Don't meet requirement (i)? Why permutation invariant? Explain the batch-wise embedding.
>
> A5-1: Sorry for the confusion. To be clear we *do* enforce requirement (i). The meta-critic input must depend on $\phi$ in order to work. Our comment here is just that the most naive way to enforce this requirement (set $input=\phi$, used in MetaReg NeurIPS'18) is slow and inaccurate. Instead we establish the dependency through a computation graph by setting $input=f(\phi,state)$, which turns out to be faster and more effective (evaluated in Tab 2).
>
> A5-2: A mini-batch is sampled from the replay buffer. Each state in the batch is embedded by MLP and then combined by sum operator Eq (6,7). Sum operator is permutation invariant. (And such a sum operation is the simplest type of set embedding). The resulting value gives the auxiliary loss for the mini-batch.
>
> Q6: TD3 min between critics.
>
> A6: Yes we know and agree. The sentence cited was not meant to be a complete description of TD3. We did already mention this aspect in related work/pg2 "TD3 .. a variant of Double Q-learning by taking the minimum value between a pair of critics". And we have emphasized "min between critics" where you pointed out in the paper.
>
> Q7: Why HalfCheetah-v2 from gym-mujoco and HalfCheetah from rllab?
>
> A7: This choice of environments was not contrived. We first evaluated the gym  environments used by TD3 papers and SAC papers. Then to test that our method was not over-fitted to choice of environment simulator, we added some additional rllab environments.
>
> Q8: You say you are using 10 million steps, but this is not always the case.
>
> A8: Thanks. Sorry for the confusion. As per SAC paper and others, we varied the number of steps evaluated according to the typical learning speed of different basic learners in different environments. 10 million steps was the maximum number used across all environments and learners. Will clarify the text.
>
> Q9: TD3 Half Cheetah-v2 collapse at 4M steps.
>
> A9: This is due to sensitivity to seeds. One seed drops to zero performance and the others continue OK. We agree that this phenomenon is interesting and important, but a full analysis is out of the scope of the current work. The take home message for now is that MC is generally less sensitive to seeds, especially compared to DDGP and TD3 (compare variance, and fewer big drops).
>
> Q10: Debatable whether SAC-MC gives a clear boost for several tasks.
>
> A10: We assess this informally from qualitative visual inspection of visible gap between the lines for several curves in Fig 4. Learning *rate* per-se is hard to quantify for reasons the reviewer mentions. Peak performance is quantified in Fig 5.
>
> Q11: Typos.
> A11: Thanks. We corrected them.

---

### Official Review · AnonReviewer2 · 2019-10-22
**Official Blind Review #2**

**Rating:** 6

**Review:**

In actor-critic algorithms, the policy tries to optimize cumulative discounted rewards by a loss formed with components from the critic. In this paper, the authors propose a novel way, namely, meta-critic, which utilizes meta-learning to learn an additional loss for the policy to accelerate the learning process of the agent. There are several advantages of the proposed method, it's learned online for a single task, it's trained with off-policy data, it provides sample efficiency and it's generally applicable to existing off-policy actor-critic methods.

Overall the proposed method is novel and the research direction is a very interesting one to explore. Eqn (3) and Eqn (4) explains the key idea of the proposed method. Eqn (3) describes the meta-learning problem as a bi-level optimization problem where the agent is updated with the main loss L^main with data d_train, in addition, it's updated with L^aux where the loss is learned and parameterized by \omega. After the agent being updated, it uses L^meta and data d_val to validate the performance of the updated agent. Eqn (4) describes an explicit way to formalize the usage of the auxiliary loss, which is to accelerate the learning process. Thus the meta loss is whether the L^aux helps the learning process or not.

Hope that the authors could address the following issues in the rebuttal:
1) Investigate why DDPG with meta-critic gets much more improvements than TD3/SAC;
2) Show SAC (or TD3) could get better performance on harder task. (I can understand that they are strong baselines and hard to improve on the current environments, however, for harder environments, there might be rooms for improvements.);
3) Investigate different ways to parameterize the meta-critic other than simple MLP;

A few minor points:
1) Page 2 first paragraph, "This is in stark contrast to the
mainstream meta-learning research paradigm – where entire task families are required to provide
enough data for meta-learning, and to provide new tasks to amortize the huge cost of meta-learning.". Try to revise it and avoid the words like "mainstream". Think about the paper being read 5 years later or even more;
2) Consider introducing a weighting hyperparam between L^main and L^aux in Eqn (3), these two losses might have different scales and it might be better to weigh them differently;
3) Minor literature detail: Page 7 "Comparison vs PPO-LIRPG" mentioned that "Intrinsic Reward Learning for PPO (Zheng et al., 2018) is the only existing online meta-learning method that we are aware of.", however, AFAIK, "Meta-Gradient Reinforcement Learning" in NeurIPS 2018 and "Discovery of Useful Questions as Auxiliary Tasks" NeurIPS 2019 are methods where meta-learning is applied online and for a single task;

**Experience Assessment:**

I have published one or two papers in this area.

**Review Assessment: Checking Correctness Of Derivations And Theory:**

I assessed the sensibility of the derivations and theory.

**Review Assessment: Checking Correctness Of Experiments:**

I assessed the sensibility of the experiments.

**Review Assessment: Thoroughness In Paper Reading:**

I read the paper at least twice and used my best judgement in assessing the paper.

---

> ### Author Response · Authors · 2019-11-13
> **Author response to Review #2**
>
> We thank the reviewer for appreciating our contribution as "novel and interesting". Below, we answer the questions raised:
>
> Q1: Why does DDPG benefit more than TD3/SAC?
>
> A1: Meta-Critic learns an additive correction factor to an existing loss. This may lead to various improvements (eg: potentially tuning exploration, entropy, or learning rate: See R1Q3). But one of the easiest augmentations to learn is a correction factor for the over-confidence that is widely known to be exhibited by DDPG. So it's easy for MC to make big improvements here. In contrast, because TD3/SAC already ameliorate this over confidence via optimizing "min(Q1,Q2)", it removes one easy and large source of improvement for MC to make over the baselines. So the margin is less.
>
> Q2: Does meta-critic provide better improvement on TD3/SAC for harder environments?
>
> A2: Thanks for the suggestion. We evaluated SAC and SAC-MC on two harder tasks, namely TORCS and Humanoid (rllab). There is a clear improvement in the results. Please see Sec 8.2 in the revised manuscript.
>
> Q3: Are there better ways to parameterize the meta-critic rather than a simple MLP?
>
> A3: We already evaluate some simple parameterization variants for the auxiliary loss in Tab 2 of the manuscript. We also considered a more sophisticated parameterization, namely that in EPG (Houthooft NeurIPS'18 Evolved Policy Gradients), which performs temporal convolutions over the agent's past experience. We tested meta-critic with this auxiliary loss parameterization on several gym-mujoco environments and found the following max returns:
>
> Environment                Vanilla DDPG.     DDPG-MC (MLP).      DDPG-MC (EPG)
> Half Cheetah:               8440.2,                 10187.5,                      1004.7
> Hopper:                         2097.5,                 3253.6,                        565.3
> Walker2D:                     2920.1,                 3753.7,                        739.7
> InvPendulum:              1000.0,                 1000.0,                         84.5
> InvDouPend:                9307.5,                 9326.5,                        113.3
>
> The overall performance order is:  DDPG-MC(MLP) > Vanilla DDPG > DDPG-MC(EPG). EPG-style parameterization does not perform well in our off-policy setting. This may be because EPG parameterization needs continuous trajectories while the simple MLP parameterization benefits from learning from random transitions in the buffer. Off-policy learning from random transitions rather is generally better due to avoiding the correlations introduced by learning from complete trajectory sequences.
>
> Q4: Minor points.
>
> A4-1: Thanks, adjusted.
> A4-2: We considered a weighting hyperparameter for the auxiliary loss. However since $L^{aux}$ is learned, its scale is also trained by the meta-loss, so it would likely be redundant to introduce another parameter. Meanwhile it would create an additional hyperparameter to potentially tune, which we wanted to avoid.
> A4-3: Thanks for pointing these out. These papers focus on learning meta-parameters (such as discount factor, bootstrapping param) online with meta-learning. We have modified our paper to update the mentioned statement, and to cite these papers. What we meant to say that PPO-LIRPG is the most similar paper to ours from the perspective of directly learning an auxiliary reward (or loss) via a neural network as an assistant to the main learning task.

---

> > ### Comment · AnonReviewer2 · 2019-11-14
> > **Feedback on Author's Response**
> >
> > I really appreciate authors' effort to respond to my concerns on the paper. The response and the revision of the paper have addressed my concerns, especially
> >
> > 1) the authors have shown in Section 8.2 Additional Environments that on harder environments, state-of-the-art off-policy actor-critic algorithms like Soft Actor-Critic (SAC) could be improved by the proposed meta-critic algorithm, which is very impressive; I'd recommend the authors to include these impressive results to the main content of the paper instead of leaving them in the Appendix;
> >
> > 2) The comparison on different meta network architecture is also very interesting.

---

### Official Review · AnonReviewer1 · 2019-10-24
**Official Blind Review #1**

**Rating:** 3

**Review:**

I am very torn about this paper as the proposed approach is a fairly straightforward extension of past work on the meta-critic approach to meta-learning and the results are pretty good, but nothing amazing. I tend to accept this paper because I like their general direction and think what they are proposing is pretty simple with broad applicability. It should be fairly straightforward to append this idea to most new off policy methods as they come out, so I find their consistent gains across 3 different popular models pretty convincing that this could have value to the community.

That being said, the gains are not huge, which does make me think about the potential computational overhead. How much more run time per step does the meta-critic add to the models in the paper? I am a bit worried that the comparisons are not apples to apples from the perspective of the amount of computation/update steps per environment interaction due to the use of the validation data. I wonder if it changes the conclusion at all if the other approaches are given the same amount of computation on their replay buffer between interactions with the environment. For example, more optimization steps on the buffer is another plausible explanation why the meta-critic does better at optimizing the loss.

I also should note that this paper is not the first to propose conducting online meta-learning over a replay buffer. A paper at last year's ICLR [1] did so in the context of lifelong learning, but does not need task labels or tasks and was tested on single non-stationary environments as well. The Meta-Critic approach of this work, of course, still is cool as it does for learning with a Meta-Critic what Meta-Experience Replay does for optimization based meta-learning. However, I thought this should be pointed out as the novelty can be a bit overstated at times. Additionally, the paper would be significantly improved by fleshing out the theoretical motivation for the Meta-Critic approach in more detail. What are the underlying reasons why we would expect it to generically improve single task RL?

[1] "Learning to Learn without Forgetting by Maximizing Transfer and Minimizing Interference". Matthew Riemer, Ignacio Cases, Robert Ajemian, Miao Liu, Irina Rish, Yuhai Tu, and Gerald Tesauro. ICLR-19.

Thoughts After Author Feedback:

I really appreciate the response of the authors to my review, which included some interesting new experiments and explanations addressing concerns I raised. I do, however, also see where reviewer 3 is coming from with both major comments.

I don't think that R3A1 is particularly clear and it is an important concern. I think reviewer 3 is saying that Delta(t) in R3A1i will eventually converge to 0 when the policy stops changing while the author argue that it "need not converge if there are fluctuations at each iteration". So it not converging seems to be tied to the existence of some source of non-stationarity in the problem. This doesn't seem to be coming from the environment as they are considering single task settings. As a result, I believe the source of the non-stationarity in this case is the fluctuating parameters. Looking at figure 6a it does not seem obvious that the policy has really converged in the traditional sense as its score does seem to be changing to some degree throughout the chart. My best guess is that this is the reason why the meta-loss does not converge. However, I still totally agree that this is a major concern that is very much under addressed.

I also agree that I found the comments about what the meta-critic is doing unconvincing. The authors provided a few different kinds of explanations of what the model could potentially be doing, but this approach to the answer really highlights  how the theoretical benefits of this approach remain unclear. I think it should be possible to directly verify some of these theories with well designed experiments. It feels like a better explanation is necessary in light of the often small margin of difference with baselines before publication.


**Experience Assessment:**

I have published in this field for several years.

**Review Assessment: Checking Correctness Of Derivations And Theory:**

I assessed the sensibility of the derivations and theory.

**Review Assessment: Checking Correctness Of Experiments:**

I assessed the sensibility of the experiments.

**Review Assessment: Thoroughness In Paper Reading:**

I read the paper at least twice and used my best judgement in assessing the paper.

---

> ### Author Response · Authors · 2019-11-13
> **Author response to Review #1 (part 1)**
>
> We thank the reviewer for appreciating our contribution as "simple and broadly applicable" providing "consistent gains". Below, we answer the questions raised:
>
> Q1: How much more run time per step does the meta-critic add to the models in the paper? .... Does it change the conclusion if the other approaches are given the same amount of computation on their replay buffer?
>
> A1: Meta-Critic takes up to 30% more computation than the baseline. The numbers below give the total cost (seconds) per step for the baseline and the meta-critic enhanced baseline, as well as the breakdown of meta-critic cost between auxiliary loss and meta-loss. The cost of evaluating the auxiliary loss cost is similar across the methods (halved for TD3 because of delayed policy update). The majority of extra cost is spent on evaluating the meta-loss, which varies across methods as it depends on the $L^{main}$, which varies across methods.
>
>                      Baseline    +MC Total    Auxiliary: Sec/%     Meta: Sec/%
> DDPG          0.007712    0.010625     0.001342s/12.6%    0.001571/14.7%
> TD3             0.008056     0.009964     0.000682s/6.22%    0.001129/10.3%
> SAC             0.02523       0.036345     0.001301s/3.57%    0.009814/27%
>
> We evaluated the impact of providing the baseline with a corresponding increase in the number of iterations. However there is no straightforward link between increasing the gradient steps per epoch, and the resulting performance of the baseline. Please see Sec 8.1 of the updated paper for details.
>
> Q2: Related Work: "Learning to Learn without Forgetting by Maximizing Transfer and Minimizing Interference".
>
> A2: Thanks for pointing out this paper. It is indeed a great related work, and we have updated our manuscript to cite it and tuned the scope of our claims accordingly. In summary, MetaExperienceReplay aims to address the transfer vs forgetting trade-off in continual learning via developing a Reptile-like algorithm in an Experience Replay setting for a continual learning. The main differences are: (1) They meta-learn parameters, while we meta-learn a loss. (2) They focus on learning multiple tasks learning, albeit in a continual way, while we focus on online learning of a single task. While their algorithm is online and could potentially be applied to a single task, the primary quantitative benefits come in the resulting cross-task transfer in a multiple task situation, as shown by their results.

---

> ### Author Response · Authors · 2019-11-13
> **Author response to Review #1 (part 2)**
>
> Q3: Flesh out theoretical motivation why Meta-Critic should improve single task RL?
>
> A3-1: At the highest level the intuition is exactly the same as the loss learner in (Houthooft NeurIPS'18 Evolved Policy Gradients; Wu NeurIPS18, Learn to Teach), and reward learner in (Zheng NeurIPS'18, Learning Intrinsic Rewards for PG). To summarize: We use "look ahead" to the outcome of learning with a loss in order, and use that outcome as a signal the loss to provide a good teaching signal for the base learner in the next iteration. This learned teaching signal could potentially help to tune various hard to tune facets of learning such as: appropriate calibration of exploration, uncertainty, or learning rate.
>
> In LIRPG, policy gradient optimizes the sum of extrinsic and intrinsic reward, while simultaneously optimising the intrinsic reward to maximize the extrinsic reward achieved by the policy using that intrinsic reward. Since the intrinsic reward is trained to produce a policy with max extrinsic reward, it indirectly assists the policy to maximise the extrinsic reward. Our meta-critic is a *loss* analogy to their intrinsic *reward*. A key contribution of ours is an online off-policy formulation, so meta-critic is an auxiliary "intrinsic Q-value" analogous to LIRPG's auxiliary "intrinsic reward". But the logic of why they work is the same: In both cases the teaching signal (auxiliary loss/intrinsic reward) is essentially updated with the same goal as the student policy (main loss/extrinsic reward). This process thus trains an auxiliary teaching signal to assist the main optimization. LIRPG's intuition is intrinsic reward learning helps exploration.
>
> EPG optimizes a loss like us, and with the same setup of optimizing the intrinsic loss, so as to maximize the extrinsic reward after the learning with that intrinsic loss. The intuition is that the intrinsic loss will end up with the same minimum as the extrinsic one, but be easier to optimize. For example, by learning to incentivise an appropriate amount of uncertainty (cf: Entropy regularizers) or appropriate amount of exploration (cf: Novelty bonuses) -- parameters which are otherwise hand-tuned. Of course EPG could be applied to a single task, and would eventually achieve a fast learning loss. But in EPG's setup, the overhead of training the policy to completion in the inner loop would mean a very low sample efficiency overall. Our contribution is showing how such auxiliary loss learning can be done online, and fast enough to provide useful assistance in single-task RL.
>
> A3-2: The specific mechanism of Meta-Critic's contribution is non-trivial to unpack as in the end it is a black box neural network. However, besides by potentially (meta)learning an appropriate amount of uncertainty/exploration (as claimed by EPG and LIRPG), it may also act as a gradient corrector/regulator. In each step the vanilla critic provides a Q-value as the main loss for the actor, then meta-critic provides an auxiliary gradient. The meta-loss asks "Did the auxiliary loss improve the performance (as measured by main loss on validation batch)?" and adjusts the auxiliary loss accordingly. This kind of loop could also potentially provide dynamic learning rate adjustment through meta-learning on validation instances from the replay buffer. See also R2Q1.

---

### Decision · Program_Chairs · 2019-12-19

**Decision:**

Reject

**Comment:**

There was extension discussion of the paper between the reviewers. It's clear that the reviewers appreciated the main idea in the paper, and the notion of an "online" meta-critic that accelerates the RL process is definitely very appealing. However, there were unanswered questions about what the method is actually doing that make me reticent to recommend acceptance at this point. I would refer the authors to R3 and R1 for an in-depth discussion of the issues, but the short summary is that it's not clear whether, if, and how the meta-loss in this case actually converges, and what the meta-critic is actually doing. In the absence of a theoretical understanding for what the modification does to accelerate RL, we are left with the empirical experiments, and there it is necessary to consider alternative hypotheses and perform detailed ablation analyses to understand that the method really works for the reasons stated by the authors (and not some of the alternative explanations, see e.g. R3). While there is nothing wrong with a result that is primarily empirical, it is important to analyze that the empirical gains really are happening for the reasons claimed, and to carefully study convergence and asymptotic properties of the algorithm. The comparatively diminished gains with the stronger algorithms (TD3 and especially SAC) make me more skeptical. Therefore, I would recommend that the paper not be accepted at this time, though I encourage the authors to resubmit with a more in-depth experimental evaluation.